# Anti-Tumor Activity of Atractylenolide I in Human Colon Adenocarcinoma In Vitro

**DOI:** 10.3390/molecules25010212

**Published:** 2020-01-04

**Authors:** Ka Woon Karen Chan, Hau Yin Chung, Wing Shing Ho

**Affiliations:** School of Life Sciences, The Chinese University of Hong Kong, Shatin, Hong Kong SAR, China; karencp600@yahoo.com.hk (K.W.K.C.); anthonychung@cuhk.edu.hk (H.Y.C.)

**Keywords:** Atractylenolide I, *Atractylodes macrocephala*, colon adenocarcinoma, HT-29, mitochondria-dependent apoptosis, caspase

## Abstract

*Atractylodes macrocephala* is known to exhibit multi-arrays of biologic activity in vitro. However, detail of its anti-tumor activity is lacking. In this study, the effects of atractylenolide I (AT-I), a bio-active compound present in *Atractylodes macrocephala* rhizome was studied in the human colorectal adenocarcinoma cell line HT-29. The results showed that AT-I induced apoptosis of human colon cancer cells through activation of the mitochondria-dependent pathway. The IC_50_ of AT-I was 277.6 μM, 95.7 μM and 57.4 μM, after 24, 48 and 72 h of incubation with HT-29, respectively. TUNEL and Annexin V-FITC/PI double stain assays showed HT-29 DNA fragmentation after cell treatment with various AT-I concentrations. Western blotting analysis revealed activation of both initiator and executioner caspases, including caspase 3, caspase 7, and caspase 9, as well as PARP, after HT-29 treatment with AT-I via downregulation of pro-survival Bcl-2, and upregulation of anti-survival Bcl-2 family proteins, including Bax, Bak, Bad, Bim, Bid and Puma. The studies show for the first time that AT-I is an effective drug candidate towards the HT-29 cell.

## 1. Introduction

Colon adenocarcinoma is the third most common malignant disease in Asia, and the fourth most common cause of cancer-related mortality globally [1]. In recent decades, the incidence of colon adenocarcinoma has increased dramatically, especially in the Asia-Pacific region, which may be due to changes in lifestyle and epigenetic factors [2,3,4]. The emergence of advanced technologies used for early screening, such as colonoscopy, effectively detect and diagnose cases of colon adenocarcinoma [5,6]. Early diagnosis, especially prior to symptom onset, significantly improves the prognosis and survival rate [7]. However, colon adenocarcinoma is often diagnosed at a late cancer stage, partly due to the lack of healthcare support and public awareness of the disease, especially in developing countries [8,9]. Advanced oncology pharmaceutical drugs are available for colon adenocarcinoma, but reoccurrence rate is high. Therefore, colon adenocarcinoma remains a major public health burden, especially in Asia-Pacific countries [10].

The majority of the existing chemotherapeutic agents are synthetic chemicals that are cytotoxic, and likely to result in a range of side effects and health complications during the treatment period [11,12]. On the contrary, natural herbal medicine treatments are generally less cytotoxic and produce fewer side effects [13]. There are over 400 species of medicinal herbs documented, offering thousands of potentially active components for treatment of a variety of diseases [14,15]. Amongst these herbal medicines, the rhizome of *Atractylodes macrocephala* Koidz, (also known as Baizhu in China) is a medicinal plant that has long been used as a tonic agent in various ethno-medical systems in Asia, and is a common, affordable herbal plants that is well-known for its anti-inflammatory effects and anti-oxidant properties, especially on the gastrointestinal tract [16,17,18]. Like most herbal medicines, it consists of various sesquiterpenes and flavones, which are believed to be the active components that give it its beneficial effects. Therefore, identification of Baixhu bioactive compounds is important for the investigation of potential medical options for various gastrointestinal disease.

Atractylenolide I (AT-I) is one of the naturally occurring compounds found in the rhizome of *Atractylodes macrocephala* [18]. It is a sesquiterpene compound, and its chemical structure is shown in Figure 1. Recent studies show that AT-I exhibits anticancer properties on various cancer cell lines, including bladder, lung and ovarian cancer [19,20,21,22]. However, no studies have investigated the effect of AT-I on human colon adenocarcinoma. It is unclear if AT-I exhibits anticancer activity on human colon adenocarcinoma. Furthermore, in the majority of the previous studies, the authors suggested that AT-I exhibits anti-tumor activity in a dose-dependent manner. In this study, the anticancer effect of AT-I on the human colorectal cancer cell line, HT-29, was analyzed in vitro, and the underlying molecular pathway involved in cancer cell death was investigated.

## 2. Results

### 2.1. Anti-Proliferative Effects of AT-I on the HT-29 Cell Line

The cell viability of the HT-29 cell line was assessed by MTT assay after the cells were treated with various concentrations of AT-1. The IC_50_ values were 277.6 μM, 95.7 μM and 57.4 μM, after AT-I treatment time of 24, 48 and 72 h, respectively. Regardless of AT-1 incubation time, the effect of AT-I treatment was most significant at 200 μM. As shown in Figure 2, an almost linear curve is presented between AT-I concentrations of 0 μM to 100 μM, suggesting cell viability decreased significantly in a concentration-dependent manner at an AT-I concentration of below 100 μM. However, the curve becomes asymptotic as the AT-I concentration exceeds 100 μM, showing similar anti-proliferative effects for concentrations exceeding 100 μM and suggesting that the maximal effective concentration is around 100 μM. AT-I was shown to possess an anti-proliferative effect on the HT-29 cell line in both a concentration- and time-dependent manner.

### 2.2. Necrotic Effects of AT-I on the HT-29 Cell Line

The lactate dehydrogenase (LDH) assay was used to assess the effect of AT-I on necrotic cell death of the HT-29 cell line. The assay measured LDH activity in cell culture medium to determine the relative level of necrosis. As shown in Figure 3, cells treated with various concentrations of AT-I for 24, 48 and 72 h displayed similar LDH activities, which were comparable to the control group. The level of LDH activity in the control group was even slightly greater than all other groups that had been treated with AT-I. Overall, this experiment suggested that treatment with AT-I did not have an effect on HT-29 necrotic cell death, regardless of AT-I concentration or treatment time.

### 2.3. Effects of AT-I on DNA Fragmentation in the HT-29 Cell Line

DNA fragmentation on the HT-29 cell line was assessed by the terminal deoxynucleotidyl transferase dUTP nick end labeling (TUNEL) assay after the cells were treated with various concentrations of AT-I for 48 h. Flourescein-12-dUTPs were used to bind with free 3′-hydroxyl ends of the fragmented DNA, which was then analyzed by flow cytometry. Figure 4a displays a gradual increase of DNA fragmentation as cells were treated with increasing concentration of AT-I. HT-29 cells that were treated with 100 μM AT-I demonstrated a significantly greater level of DNA fragmentation. The experiment was repeated three times and the average apoptotic index level was calculated. After treatment with AT-I at 10, 20, 40, 80 and 100 μM, the mean apoptotic index was 2.94, 6.26, 8.64, 19.50 and 23.79%, respectively, in comparison with the control group of 1.94% (Figure 4b). The result suggested that treatment with AT-I may induce DNA fragmentation in the HT-29 cell line in a dose-dependent manner.

### 2.4. Apoptosis Induction of the HT-29 Cell Line by AT-I

Apoptosis levels of HT-29 cell line were assessed by Annexin V-FITC/PI double stain using flow cytometry, after cells were treated with various concentrations of AT-I for 48 h. Annexin V conjugated to FITC fluorescence dye binds to phosphatidylserine on apoptotic cells, whereas propidium iodide (PI) binds to necrotic cells. Figure 5a shows the increased number of cells in the late apoptosis phase (top right quadrant) as cells were treated with higher concentrations of AT-I. HT-29 cells that were treated with 100 μM AT-I had the highest number of apoptotic cells. The experiment was repeated three times and the average number of cells at the late and early apoptosis phase were determined, as shown in Figure 5b. After treatment with AT-I at 10, 20, 40, 80 and 100 μM, the mean percentage of cells at the late apoptosis phase were 10.56, 12.83, 16.54, 45.56 and 54.28%, respectively, in comparison with the control group of 8.82%. The mean percentage of cells at early apoptosis were 5.79, 5.32, 7.62, 18.14 and 19.64%, respectively, in comparison with the control group of 4.82%. The number of cells at the late apoptosis phase were considerably greater than the number of cells at the early apoptosis phase. The result suggested that treatment with AT-I may induce apoptosis—and more significantly late apoptosis—in the HT-29 cell line in a dose-dependent manner.

### 2.5. The effect of AT-I on the Expression Level of Caspases in the HT-29 Cell Line

Western blotting was used to determine the expression level of various proteins that play essential roles in programmed cell death. In particular, expression levels of both initiator and executioner caspases were analyzed, after cells were treated with various concentrations of AT-I for 48 h. The total protein concentration of each sample was measured using the Bradford protein assay and normalized. β-tubulin was probed to ensure an equivalent amount of sample protein was loaded into each lane. Figure 6a is a representative of three experiments and demonstrates the protein expression level of activated apoptotic proteins—caspase 9, caspase 3 and caspase 7, as well as poly ADP-ribose polymerase (PARP) after HT-29 cell treatment with AT-I for 48 h. Furthermore, the overall expression levels of pro-caspase 9, pro-caspase 3, pro-caspase 7, and pro-PARP were shown to be reduced. Figure 6b presents the percentage ratio of each investigated protein to β-tubulin, after treatment with 10, 20, 40, 80 and 100 μM AT-I for 48 h. After treatment with 100 μM AT-I, the HT-29 cell line protein expression level of cleaved caspase 9, cleaved caspase 3, cleaved caspase 7 and cleaved PARP were increased by 265.8, 391.3, 478.9 and 658.8%, respectively, and in comparison, with the control group (100%). Overall, western blotting analysis indicated that AT-I may increase expression levels of various caspases in a dose-dependent manner and suggested that AT-I may be able to induce caspase-dependent apoptosis in the HT-29 cell line.

### 2.6. The effect of AT-I on the Expression Level of Caspase 8 in HT-29 Cell Line

To investigate the effect of AT-I on extrinsic apoptosis, the expression level of caspase 8 was measured using western blotting, after treatment with various concentrations of AT-I for 48 h. Figure 7a is a representative of three individual experiments; it shows that the band intensity increased in a concentration-dependent manner. A significant increase in caspase 8 expression level was seen after treatment with AT-I at a concentration of 40 μM or greater. As shown in Figure 7b, the percentage ratio of cleaved caspase 8 versus β-tubulin, after treatment with 10, 20, 40, 80 and 100 μM AT-I, increased by 237.1, 245.7, 431.4, 548.7 and 775.7%, respectively, in comparison to the control group (100%). The result indicated that AT-I may increase expression levels of cleaved caspase 8 in a dose-dependent manner and suggested that AT-I may be able to induce extrinsic apoptosis in the HT-29 cell line.

### 2.7. The Effect of AT-I on the Expression Level of the Pro-Apoptotic Bcl-2 Family Proteins in the HT-29 Cell Line

To investigate the effect of AT-I on mitochondria-dependent apoptosis, the expression level of the pro-apoptotic Bcl-2 family of proteins was measured using western blotting, after treatment with various concentrations of AT-I for 48 h. Figure 8a is a representative of three individual experiments; it shows that the band intensity of all assessed proteins increased in a concentration-dependent manner, while the band intensity of the loading control—β-tubulin—remained constant. Figure 8b presents the percentage ratio of the assessed proteins versus β-tubulin. After treatment with 100 μM AT-I, the HT-29 cell line protein expression level of Bax, Bak, Bad, Puma and Bid were significantly increased by 290.8, 438.5, 194.3, 608.9 and 223.6%, respectively, in comparison with the control group (100%). On the contrary, the expression level of Bim was not markedly altered by AT-I treatment, although a slight increase was observed at 40 μM and above. The experiment demonstrated that AT-I treatment with HT-29 cell line for 48 h may result in an increased expression level of the pro-apoptotic Bcl-2 family in a concentration-dependent manner, and suggested that AT-I was able to induce mitochondria-dependent apoptosis.

### 2.8. The Effect of AT-I on the Expression Level of the Pro-Survival Bcl-2 Family of Proteins in the HT-29 Cell Line

In addition to pro-apoptotic Bcl-2 family proteins, the expression level of pro-survival Bcl-2 family proteins, Bcl-2 and Bcl-xL, were also analyzed after treatment with various concentrations of AT-I for 48 h. Figure 9a is a representative of three individual experiments; it shows the band intensity of Bcl-2 and Bcl-xL from the western blot experiment. Figure 9b presents the percentage ratio of Bcl-2 and Bcl-xL versus β-tubulin. The expression level of Bcl-2 decreased to 64.9%, 35.9%, 22.8%, 14.1% and 19.8% of the control after treatment with 10, 20, 40, 80 and 100 μM, respectively. The expression level of Bcl-xL changed to 97.8%, 110.4%, 106.0%, 94.5% and 97.3% of the control after treatment with 10, 20, 40, 80 and 100 μM, respectively. The experiment demonstrated that after treatment with AT-I for 48 h, the expression level of Bcl-2 decreased significantly in a concentration dependent manner, whereas the expression level of Bcl-xL remained unchanged regardless of AT-I concentration.

## 3. Discussion

Although the medicinal properties of *Atractylodes macrocephala* are known, the anti-tumor activity of *Atractylodes macrocephala* remains unclear. Previous studies suggested that AT-I exhibits anti-tumor effects on cancer cell lines but its effect on colon carcinoma is not fully understood.

This study is the first time that the anti-tumor activity of AT-I in human colon cancer cells through the mitochondria-dependent pathway has been shown.

The present study suggested that AT-I possesses an anti-proliferative effect on the HT-29 cell line, which is inconsistent with previous studies that showed that AT-I exhibits an inhibitory effect on human cell lines in vitro [19,20,21,22]. The maximum effective dose was found to be 200 μM. The TUNEL assay provided supportive experimental evidence on the anti-tumor activity of AT-I, while the LDH assay showed that AT-I had little effect on HT-29 necrotic cell death. Flow cytometry using Annexin-V FITC/PI double stain revealed the apoptosis level of HT-29 cells after treatment with various concentrations of AT-1.

Western blotting showed an increased expression level of the activated apoptotic proteins, caspase 9, caspase 3 and caspase 7, as well as poly ADP-ribose polymerase (PARP). Caspase 9 is an initiator caspase, a monomeric zymogen, which when activated, triggers activation of executioner caspases such as caspase 3 and 7 [23]. Activated caspase 3 is an aspartate-specific cysteine protease that triggers cleavage of PARP and onset of DNA fragmentation [24]. Of all the evaluated proteins, the expression level of activated caspase 9, being an initiator caspase, increased less significantly (by 265.8% compared with the control group), whereas the expression level of cleaved PARP, being a cellular substrate of caspases, increased most significantly (by 658.8% compared with the control group). Furthermore, a significant increase of cleaved caspase 8 protein (up to 775.7% of the control group) was detected by western blotting, after HT-29 cell treatment with AT-I in a dose-dependent manner. Cleavage of executioner caspases often triggers autoactivation of caspase 8, which further propagates apoptotic signals via direct activation of the pro-apoptotic Bcl-2 family of proteins, especially through Bid [25,26].

Bcl-2 family proteins, including BH3-only members (such as Bid, Bad, Bim and Puma) and multi-domain members (such as Bax and Bak), are critical components involved in mitochondrial apoptosis. Pro-apoptotic Bcl-2 family proteins interact with mitochondria, induce mitochondrial outer member permeabilization and trigger mitochondria-dependent apoptosis. Western blotting demonstrated increased expression level of the pro-apoptotic Bcl-2 family of proteins (Bax, Bak, Bad, Puma and Bid) after AT-I treatment on the HT-29 cell line. On the contrary, using the same western blotting method, the pro-survival Bcl-2 protein was found to be decreased by as much as 80% at 100 μM of AT-1. The most significant cellular change for treated HT-29 cells occurred at 100 μM AT-1.

In conclusion, the results demonstrated that AT-I was an effective anti-proliferative agent towards human colorectal adenocarcinoma in vitro through the activation of caspases and pro-apoptotic Bcl-2 family proteins, and that the mitochondrial-dependent pathway is involved.

## 4. Materials and Methods

### 4.1. Materials and Chemical Reagents

RPMI-1640 medium, fetal bovine serum, penicillin, streptomycin and trypsin were purchased from Gibco BRL (Grand Island, NY, USA). Atractylenolide I, MTT [3-(4, 5-dimethylthiazol-2-yl)-2,5-diphenyl tetrazolium bromide] were purchased from Sigma (St. Louis, MO, USA). De-ionized water (ddH_2_O) was collected from Millipore water purification system (Millipore, Milford, MA, USA). All rabbit polyclonal antibodies were purchased from Cell Signaling Technology (Dancers, MA, USA), except anti-caspase 8 antibody, which was purchased from Santa Cruz (Dallas, TX, USA).

### 4.2. Cell Line and Cell Culture

The human colorectal adenocarcinoma cell line HT-29 was obtained from ATCC (Manassaa, VA, USA) and cultured in RPMI 1640 medium supplemented with 10% fetal bovine serum and 1% penicillin streptomycin solution at 37 °C in a humidified atmosphere of 5% CO_2_. HT-29 cells were seeded into 96-well plates (8 × 10^3^ cells/well) and 6-well plates (1 × 10^6^/well) for experiments. Cells were treated with a pre-determined concentration of atractylenolide I in 100 μL medium for an appropriate time.

### 4.3. Cell Viability Evaluation by MTT Assay

Cell proliferation activity was measured with the MTT assay. After treatment, 10 μL of MTT (5 mg/mL) solution was added into each well before incubation at 37 °C for 4 h. All medium was removed and 150 μL of DMSO was added before absorbance measurement at 540 nm was recorded by a micro-titer plate reader (TECAN infinite M200; Molecular Devices, Waitham, MA, USA). The cell viability index was calculated according to the manufacturer’s protocol, which is experimental OD value/control OD value × 100%.

### 4.4. Cell Necrosis Evaluation by LDH Assay

LDH activity was analyzed using the LDH cytotoxicity assay kit (Promega, Waitham, MA, USA). Absorbance at 490 nm was measured by the micro-titer plate reader (TECAN infinite M200; Molecular Devices, Waitham, MA, USA). Maximum LDH activity (positive control) was achieved by addition of 10 μL lysis buffer to the medium in the positive control well 30 min prior to the experiment, and background values of treated cells were measured and subtracted from the sample reading. Necrotic cell death was calculated by comparing the percentage difference between treated cells and positive control cells.

### 4.5. DNA Fragmentation Analysis by TUNEL Assay

DNA fragmentation was analyzed with the ApoDIRECT in situ DNA fragmentation assay kit (BioVision, Milpitas, CA, USA). Treated cells were trypsinized, collected in a 15 mL falcon and centrifuged at 600× *g* for 5 min. Cells were fixed in 1 mL of 1% (*w*/*v*) paraformaldehyde in PBS and incubated on ice for 15 min. The cells were centrifuged and washed in 1 mL PBS twice. Fixed cells were added to 5 mL of ice-cold 70% (*v*/*v*) ethanol and incubated on ice for 30 min. Cells were collected by centrifugation, washed with wash buffer and incubated in 50 μL of staining solution according to the manufacturer’s protocol. Cells were rinsed in rinse buffer and incubated with PI/RNase A solution for 30 min in dark at room temperature. Fluorescence-activated cells were analyzed with FACSVerse System (BD Biosciences, San Jose, CA, USA) and 10,000 events were recorded per sample.

### 4.6. Cell Apoptosis Quantification by Annexin V-FITC/PI Double Staining Assay

Cell apoptosis was analyzed with the FITC Annexin V apoptosis detection kit I (BD Pharmingen, San Diego, CA, USA). Treated cells were trypsinized, collected in a 15 mL falcon and centrifuged at 600× *g* for 5 min. Cell pellets were washed with 1 mL 1× PBS, centrifuged at 600× *g* for 5 min. Cell pellets were re-suspended in 500 μL of 1× annexin-V binding buffer and stained with 5 μL PI and 5 μL GFP-Annexin-V for 15 min in the dark. Fluorescence-activated cells were analyzed with FACSVerse System (BD Biosciences) and 10,000 events were recorded per sample.

### 4.7. Caspase 3/7 Activity Assay

The activity of caspase 3/7 was determined with the caspase-glo 3/7 assay kit (Promega, Madison, WI, USA). After treatment, the medium was shaken well after addition of 100 μL of caspase 3/7 reagent and incubated at room temperature in the dark for 5 h. Fluorescence was recorded using a fluorescence microplate reader (TECAN infinite M200, Waitham, MA, USA) at an excitation wavelength of 485 nm and an emission wavelength of 530 nm. Caspase 3/7 activity was calculated according to the manufacturer’s protocol, which is experimental caspase3/7 value/control caspase3/7 value × 100%.

### 4.8. Western Blot Analysis

Cells were collected by trypsinization post-treatment, and lysed with 50 μL NP40 lysis buffer on ice for 1 h. Cell lysate was centrifuged at 600× *g* for 10 min at 4 °C, and the supernatant was collected and normalized with Bradford assay (Bio-Rad, Hercules, CA, USA) according to manufacturer’s protocols. An equivalent amount of SDS-loading dye was added to the normalized proteins and the mixture was boiled for 15 min. Twenty micrograms of protein from each sample was loaded, run on a 10% SDS-PAGE gels and transferred to 0.25 μm PVDF membranes (Millipore, Bedford, MA, USA). After blocking membranes in 5% non-fat milk in Tris-buffered saline with 0.1% Tween-20 (TBST) for 1 h at room temperature, the membranes were washed (three times in TBST for 5 min) and incubated with specific primary antibody (antibody dilution according to product data sheet) at 4 °C overnight. The membranes were washed and incubated with horseradish peroxide-conjugated secondary antibody at a 1:2000 dilution for 1 h at room temperature. The chemiluminescence of the membrane was measured by ECL chemiluminescence detection reagent (GE, Fairfield, CT, USA). β-tubulin was probed to ensure protein normalization.

### 4.9. Statistical Analysis

Data are expressed as mean ± SD of independent experiments (*n* ≥ 3). All control groups used in this study were vehicle controls with the same DMSO concentration in the medium (0.5% *v*/*v*) as the treatment groups. GraphPad Prism (San Diego, CA, USA) statistics software was used to perform statistical analysis. The non-linear regression test was applied to obtain a fit curve. Differences between the treatment groups and the control group were analyzed using one-way analysis of variance (ANOVA) or two-way ANOVA.

## Figures and Tables

**Figure 1 molecules-25-00212-f001:**
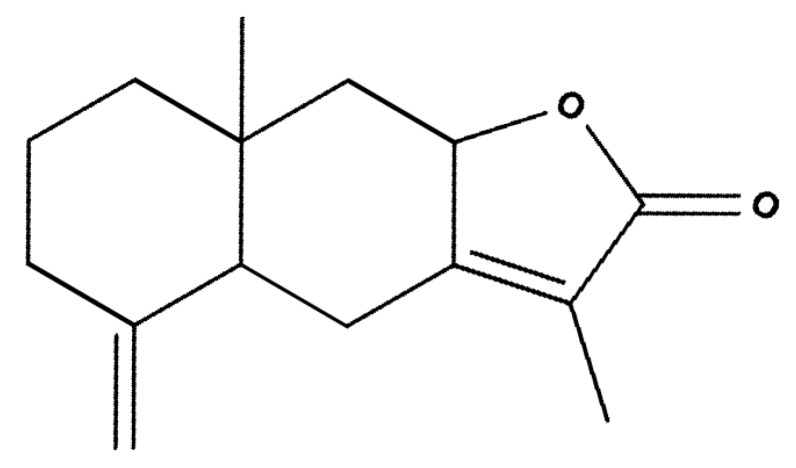
The chemical structure of atractylenolide I.

**Figure 2 molecules-25-00212-f002:**
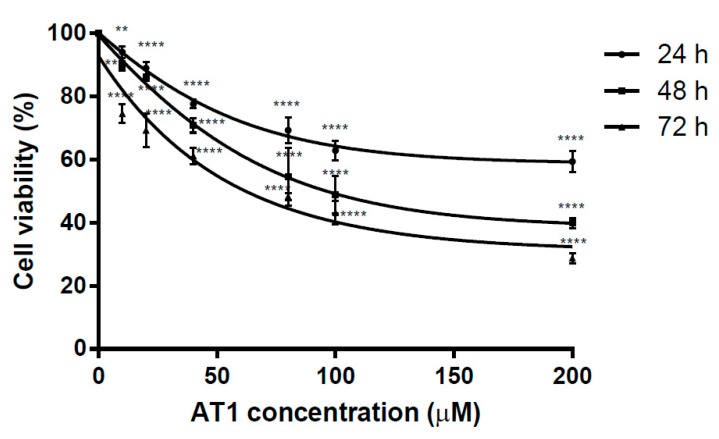
The anti-proliferative effect of various AT-1 concentrations on the HT-29 cell line, after treatment for 24, 48 and 72 h. Data are shown as mean ± SD, *n* = 6. Significant differences are indicated by ** *p* < 0.01 and **** *p* < 0.0001 as compared with the control group.

**Figure 3 molecules-25-00212-f003:**
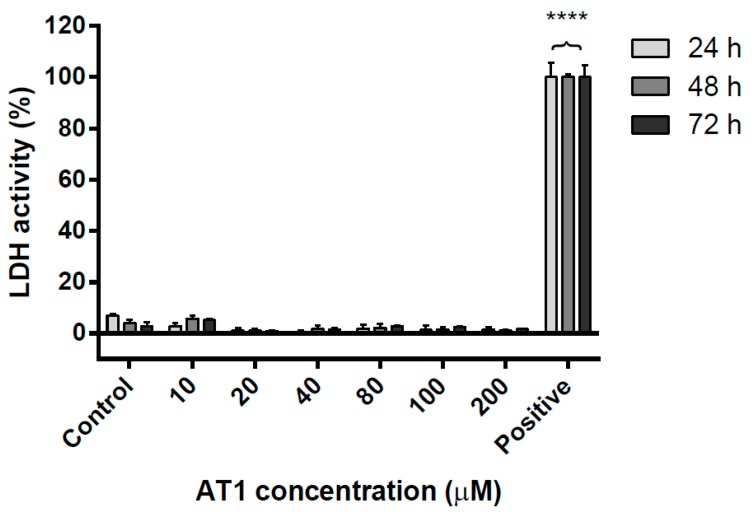
The necrotic effects of various AT-I concentrations on the HT-29 cell line, after treatment for 24, 48 and 72 h, as determined by LDH activity in cell culture medium. The control group was not treated with AT-I, showing little LDH activity. Data are shown as mean ± SD, *n* = 6. Significant differences are indicated by **** *p* < 0.0001 as compared with the control group.

**Figure 4 molecules-25-00212-f004:**
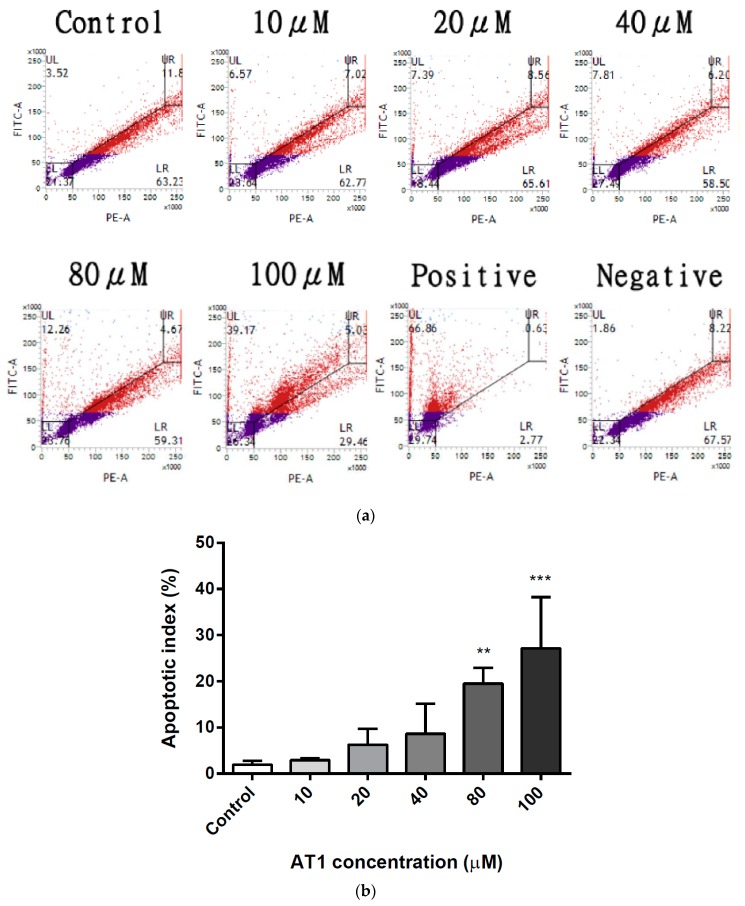
The effects of various concentrations of AT-I on apoptotic DNA fragmentation in HT-29 cells were assessed using the TUNEL assay and flow cytometry. Cells were treated with AT-I for 48 h and analyzed using flow cytometry. The control group was not treated with AT-I. (**a**) A dot plot to represent the number of events with fragmented DNA after treatment with various AT-I concentrations, as well as positive and negative controls. This is a representative of three independent experiments. (**b**) A bar chart to present the apoptotic index after treatment with various AT-I concentrations. Data are shown as mean ± SD, *n* = 3. Significant differences are indicated by ** *p* < 0.01 and *** *p* < 0.001 as compared with the control group.

**Figure 5 molecules-25-00212-f005:**
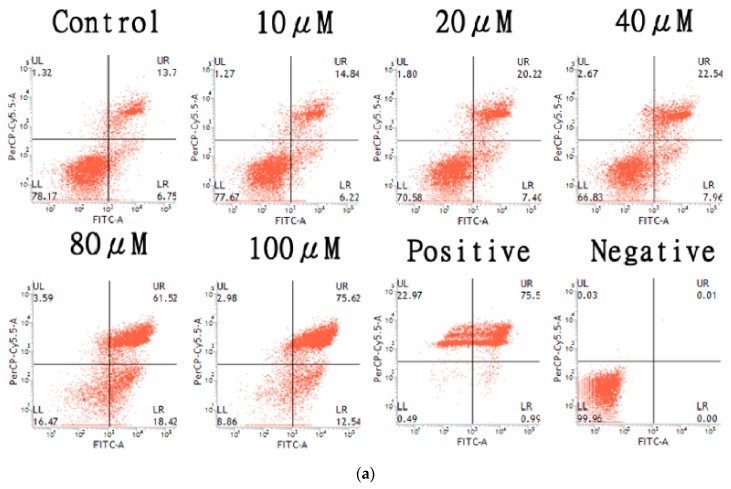
The effects of various concentrations of AT-I on apoptosis in HT-29 cells were assessed by Annexin V-FITC/PI double stain using flow cytometry. Cells were treated with AT-I for 48 h and analyzed using flow cytometry. The control group was not treated with AT-I. (**a**) A dot plot to represent number of apoptotic events after treatment with various AT-I concentrations, as well as positive and negative controls. This is a representative of three independent experiments. The top left quadrant represents dead cells (PI positive; Annexin V-FITC negative); the bottom left quadrant represents living cells (PI negative; Annexin V-FITC negative); the top right quadrant represents cells in late apoptosis phase (PI positive; Annexin V-FITC positive); the bottom right quadrant represents cells in early apoptosis phase (PI negative; Annexin V-FITC positive). (**b**) A bar chart to present the percentage of apoptotic cells after treatment with various AT-I concentrations. Data are shown as mean ± SD, *n* = 3. Significant differences are indicated by *** *p* < 0.001 as compared with the control group.

**Figure 6 molecules-25-00212-f006:**
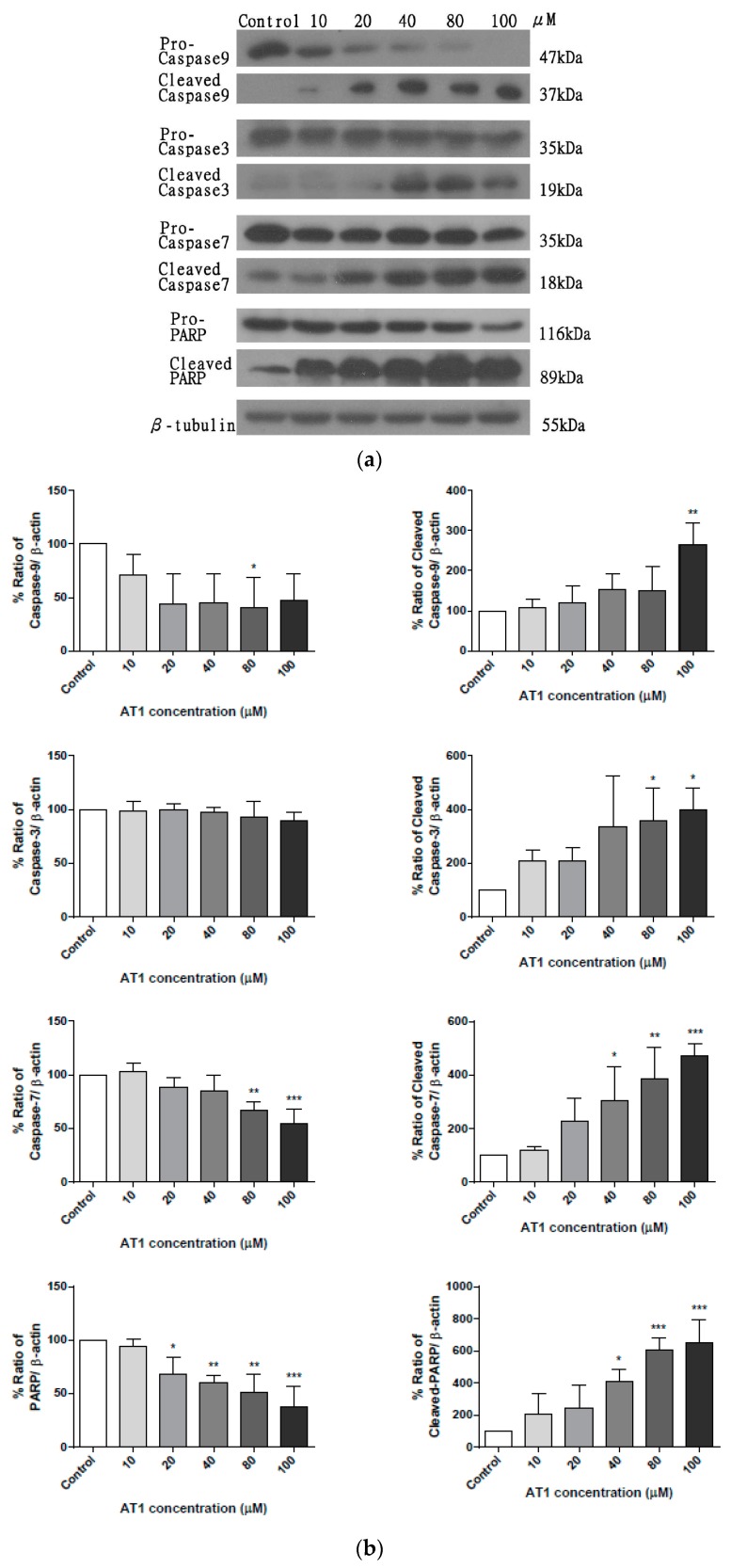
The effects of various concentrations of AT-I on the expression levels of caspases and PARP in HT-29 cells were assessed by western blotting, after cells were treated with AT-I for 48 h. The control group was not treated with AT-I. Total protein concentration of each sample was determined and normalized. β-tubulin was used as a loading control in all western blot experiments. The intensity of each band was quantified by ImageJ image processing program. (**a**) Western blot images of pro-caspase 9, cleaved capase 9, pro-caspase 3, cleaved caspase 3, pro-caspase 7, cleaved caspase 7, pro-PARP, cleaved PARP, and β-tubulin. The images are representative of three independent experiments. (**b**) Each bar chart presents the percentage ratio of each investigated protein to β-tubulin. Data are shown as mean ± SD, *n* = 3. Significant differences are indicated by * *p* < 0.1, ** *p* < 0.01, *** *p* < 0.001 as compared with the control group.

**Figure 7 molecules-25-00212-f007:**
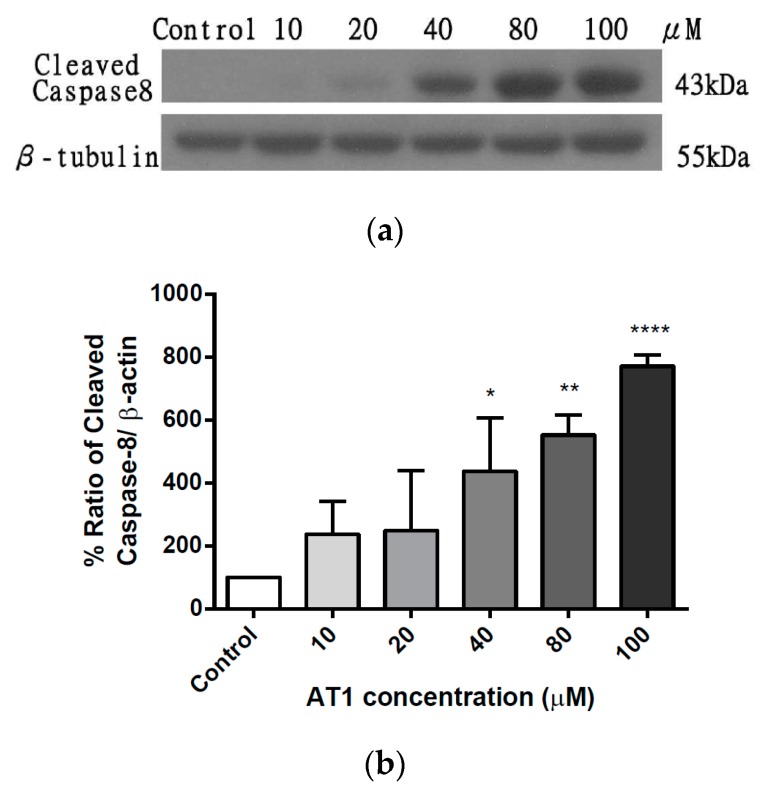
The effects of various concentrations of AT-I on the expression levels of cleaved caspase 8 was assessed by western blotting, after cells were treated with AT-I for 48 h. The control group was not treated with AT-I. Total protein concentration of each sample was determined and normalized. β-tubulin was used as a loading control. The intensity of each band was quantified by ImageJ image processing program. (**a**) Western blot images of cleaved caspase 8 and β-tubulin. The images are representative of three independent experiments. (**b**) A bar chart presents the percentage ratio of cleaved caspase 8 to β-tubulin. Data are shown as mean ± SD, *n* = 3. Significant differences are indicated by * *p* < 0.1, ** *p* < 0.01, and **** *p* < 0.0001 as compared with the control group.

**Figure 8 molecules-25-00212-f008:**
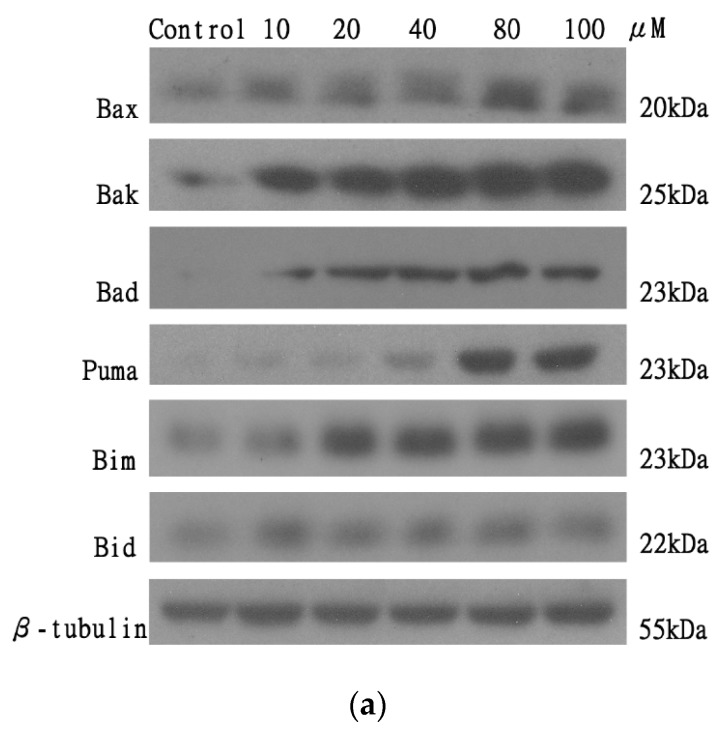
The effects of various concentrations of AT-I on the expression levels of the pro-apoptotic Bcl-2 family of proteins in HT-29 cells were assessed by western blotting, after cells were treated with AT-I for 48 h. The control group was not treated with AT-I. Total protein concentration of each sample was determined and normalized. β-tubulin was used as a loading control in all western blot experiments. The intensity of each band was quantified by ImageJ image processing program. (**a**) Western blot images of Bax, Bak, Bad, Puma, Bim, Bid and β-tubulin. The images are representative of three independent experiments. (**b**) Each bar chart presents the percentage ratio of each investigated protein to β-tubulin. Data are shown as mean ± SD, *n* = 3. Significant differences are indicated by * *p* < 0.1, ** *p* < 0.01 and *** *p* < 0.001 as compared with the control group.

**Figure 9 molecules-25-00212-f009:**
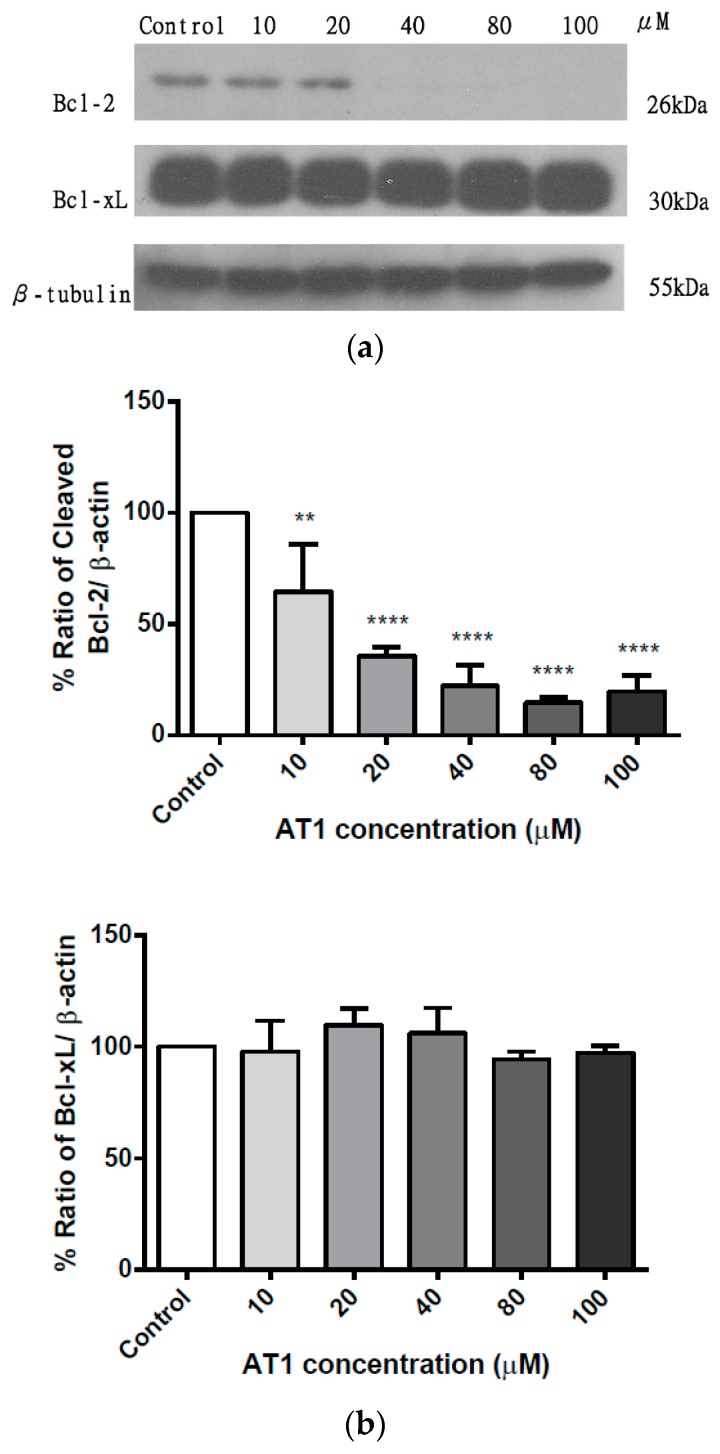
The effects of various concentrations of AT-I on the expression levels of pro-survival Bcl-2 family proteins in HT-29 cells were assessed by western blotting, after cells were treated with AT-I for 48 h. The control group was not treated with AT-I. The total protein concentration of each sample was determined and normalized. β-tubulin was used as a loading control in all western blot experiments. The intensity of each band was quantified by the ImageJ image processing program. (**a**) Western blot images of Bcl-2, Bcl-xL, and β-tubulin. The images are representative of three independent experiments. (**b**) Each bar chart presents the percentage ratio of each investigated protein to β-tubulin. Data are shown as mean ± SD, *n* = 3. Significant differences are indicated by ** *p* < 0.01 and **** *p* < 0.0001 as compared with the control group.

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
