# Peer review of "Anti-Tumor Activity of Atractylenolide I in Human Colon Adenocarcinoma In Vitro"

_molecules, 2020, doi:10.3390/molecules25010212_

Round 1
Reviewer 1 Report
Well done, indeed. Long story short, I really liked Your valuable manuscript (MS). My sincere congrats to all three authors.
Significance of Content - High; Quality of Presentation - High; Scientific Soundness - High; Overall Merit - High
Recommendation: Accept after minor revision
English language and style are fine/minor spell check required. Please, kindly improve a bit the language, if You are in position to act in such a way.
In addition to this, the authors are affably requested to consider the citing of the following two references throughout the text of their promising MS: - Arabian Journal of Chemistry 2017, Volume 10, Supplement 1, Pages S1240-S1242 - Natural Product Research 2016, Volume 30, Issue 11, Pages 1293-1296
At least in my humble opinion, this MS has a real potential to be frequently (i.e. quite well) cited (in terms of its hetero-citations), once when published.
Last but not least, very best of (research) luck ahead to You all.
Author Response
Thank you very much for the comments. As suggested, the references no. 17 and 20 have been replaced by the following two references:
17. - Arabian Journal of Chemistry 2017, Volume 10, Supplement 1, Pages S1240-S1242
20. - Natural Product Research 2016, Volume 30, Issue 11, Pages 1293-1296
And the English has been checked for correctness.
Reviewer 2 Report
The search for new compounds with anti-cancer properties is a current scientific direction.
The authors in the introduction of the article present in detail the subject of their research, describe the importance of Atractylodes macrocephaly in Chinese medicine and characterize the active component of the plant - atractylenolide I. They draw attention to the problem of colon cancer incidence in the world and in Asian countries.
The selection of research methods corresponds to the selected objectives.
It should be emphasized that this work for the first time describes the proapoptotic properties of atractylenolide I in HT-29 cell line (colon adenocarcinoma). There are only few reports on the anti-cancer properties of atractylenolide I and they cover 20-30 papers total, including an article in Molecules from 2013. All the papers have been included in the references by the authors. Therefore I think that this article should be accepted for printing in Molecules after considering the following comments:
Language - language correction required (grammar and spelling) Figure descriptions - too extensive, reproducing the content from the “Results” section. The corresponding description in Fig. 2 Materials and Methods
The data on reagents used should be completed according to the scheme (city, state, country). Correct entry has line 264: Gibco BRL (Grand Island, NY, USA)
Author Response
Thank you very much for the comments.
Gibco BRL (Grand Island, NY, USA) has been revised on line 276 as suggested.
However, the figure description add clarity to the results and may help clarify the significance of the results.
We have double checked the correctness of the English of the manuscript.
We think the figure descriptions will add clarity to the experimental setup and results. Hope readers can get a better picture of the results.
Reviewer 3 Report
The authors were studied the effects of atractylenolide I (AT-9 I), sesquiterpene lactone present in Atractylodes macrocephala rhizome on human colorectal adenocarcinomas cell line HT-29 and possible mechanisms of actions.
The authors should make some changes in the text:
give complete botanical name of the plant: plant species, authors and family. the introduction should be completed with more information about previous studies and why they selected only this cell line for testing. Is it connected with traditional use of the plant? also, the discussion should be supplemented with compairing the results with the findings of other authors, especially if there are similar or not results for other sesquiterpene lactones.
Line 2 Anti-tumor activity of not italic, Atractylenolide I – a small
Line 8 Atractylodes macrocephala – M always small
Line 9 the effects not italic, atractylenolide I , A small
Line 13 space between the number and units
Line 38 not medical-medicinal plants
Line 42-43 Give references and more information about chemistry and pharmacological activity.
References should be checked.
Author Response
Thank you very much for the comments.
The botanical name has been inserted on lines 41-42 with more relevant references included in the reference nos. 17 and 20.
This is the first detailed report that showed the mechanistic actions of AT-I in colon cancer cells. The "Introduction" clearly showed the anti-tumor activity of AT-I of this medicinal plant remains unclear until the present study.
The following comments have been well taken and corrected as suggested.
Line 2 Anti-tumor activity of not italic, Atractylenolide I – a small...............corrected
Line 8 Atractylodes macrocephala – M always small....corrected
Line 9 the effects not italic, atractylenolide I , A small....corrected
Line 13 space between the number and units..... corrected
Line 38 not medical-medicinal plants..... corrected
Line 42-43 Give references and more information about chemistry and pharmacological activity...... references added